# Comparison of MRI Visualization Following Minimally Invasive and Open TLIF: A Retrospective Single-Center Study

**DOI:** 10.3390/diagnostics11050906

**Published:** 2021-05-19

**Authors:** Vadim A. Byvaltsev, Andrei A. Kalinin, Morgan B. Giers, Valerii V. Shepelev, Yurii Ya. Pestryakov, Mikhail Yu. Biryuchkov

**Affiliations:** 1Department of Neurosurgery, Irkutsk State Medical University, 664003 Irkutsk, Russia; andrei_doc_v@mail.ru (A.A.K.); shepelev.dok@mail.ru (V.V.S.); pestryakov-nho@mail.ru (Y.Y.P.); 2Department of Neurosurgery, Railway Clinical Hospital, 664005 Irkutsk, Russia; 3School of Chemical, Biological, and Environmental Engineering, Oregon State University, Corvallis, OR 97331, USA; morgan.giers@gmail.com; 4Department of Neurosurgery with the Course of Traumatology, West Kazakhstan Marat Ospanov Medical University, Aktobe 030012, Kazakhstan; biryuchkov@mail.ru

**Keywords:** lumbar spine, degenerative diseases, transforaminal interbody fusion, open transpedicular fixation, minimally invasive decompression and stabilization, postoperative imaging, artifacts, magnetic resonance imaging

## Abstract

Analysis of magnetic resonance image (MRI) quality after open (Op)-transforaminal interbody fusion (TLIF) and minimally invasive (MI)-TLIF with the implantation of structurally different systems has not previously been performed. The objective of this study was to conduct a comparative analysis of the postoperative MRI following MI and Op one-segment TLIF. Material and Methods: The nonrandomized retrospective single-center study included 80 patients (46 men and 24 women) aged 48 + 14.2 years. In group I (*n* = 20) Op-TLIF with open transpedicular screw fixation (TSF) was performed, in II group (*n* = 60), the MI-TLIF technique was used: IIa (*n* = 20)—rigid interspinous stabilizer; IIb (*n* = 20)—unilateral TSF and contralateral facet fixation; IIc (*n* = 20)—bilateral TSF. Results: Comparison of the quality of postoperative imaging in IIa and IIb subgroups showed fewer MRI artifacts and a significantly greater MR deterioration after Op and MI TSF. Comparison of the multifidus muscle area showed less atrophy after MI-TLIF and significantly greater atrophy after Op-TLIF. Conclusion: MI-TLIF and Op-TLIF with TSF have comparable postoperative MR artifacts at the operative level, with a greater degree of muscle atrophy using the Op-TLIF. Rigid interspinous implant and unilateral TSF with contralateral facet fixation have less artifacts and changes in the multifidus muscle area.

## 1. Introduction

Transforaminal interbody fusion (TLIF) and open (Op) dorsal rigid stabilization are the most common methods of surgical treatment for most degenerative diseases of the lumbar spine [1,2]. At the same time, minimally invasive (MI) decompression and stabilization techniques are actively used in modern spinal surgery and associated with better clinical outcomes due to less paravertebral tissue damage, minimum postoperative pain syndrome, and shorter recovery [3,4]. Adverse consequences of such interventions can be cage migration, instability of the structure, degeneration of the adjacent level, and postoperative muscle atrophy; for these reasons, complete visualization of the operated and adjacent levels is a mandatory requirement for medical implants [5,6]. Even in cases where a patient has complete functional recovery, imaging is necessary following instrumental fixation to assess the degree of nerve decompression, study the state of adjacent segments and paravertebral muscles, and assess the interbody fusion formation [7]. In addition to computed tomography (CT), magnetic resonance imaging (MRI) is a required informative noninvasive method for postoperative examination of neural structures [8].

Most implants used in spinal surgery have properties that make them susceptible to artifacts in postoperative images [9]. The use of various additional software options for MRI improves visualization quality but does not eliminate artifacts [10]. This is one of the incentives for developing and implementing constructs with potentially less distortion of postoperative MR images [11].

A comparative analysis of MRI artifacts after Op and MI one-segment decompression and stabilization interventions with implantation of structurally different systems has not been performed before. Here, we assess whether the size and location of artifacts created by various implants render portions of the postoperative MRI useless and whether there are comparable procedures that allow for better postoperative imaging assessment of the patient.

The objective of this study was to conduct a comparative analysis of the postoperative MRI following minimally invasive and open one-segment TLIF.

## 2. Materials and Methods

All surgeries were performed at the Center for Neurosurgery of the Railway Clinical Hospital, a private healthcare institution in Irkutsk, Russia, between 2009 and 2019; Figure 1 presents the flow chart of the study design. The timeframes of the Op and MI surgeries were similar. This study was approved by the ethics committee at the Irkutsk State Medical University.

A random sample of patients was obtained by simple random sampling from the general database of patients (*n* = 3989) in Excel. Analysis of the postsurgical imaging was conducted in January 2021. This comparative, retrospective, single-center study included 80 patients (46 men and 24 women) aged 48 ± 14.2 years. A comprehensive instrumental analysis of MRI, CT, and lumbar X-ray before surgery and within 5 to 8 years following surgery (average follow-up history was 6.3 years) was performed in all patients.

### 2.1. Patient Inclusion Criteria

Eighty patients were nonrandomly selected from 5518 patients having received spinal decompression and stabilization surgeries at the study institution between 2009 and 2019. Patients were candidates for study inclusion if they met the following criteria:single-level degenerative disc disease of L4–L5;received TLIF and dorsal rigid stabilization performed for degenerative disc disease with foraminal stenosis or segmental instability;patient information (MRI data) was available in the follow-up.

### 2.2. Patient Exclusion Criteria

Patients were not considered for the study if they had any one of the following criteria:two-level degenerative disc disease of the lumbar spine;operative level was L1–L2/L2–L3/L3–L4/L5–S1;anterior or lateral lumbar interbody fusion with dorsal rigid stabilization;previously performed operations at the lumbar level;inability to conduct MRI in patients in the postoperative period (fear of confined spaces, the presence of foreign metal objects, etc.);competing pathological process in the lumbar spine (traumatic injuries, systemic diseases of connective tissue, infectious and inflammatory diseases, tumor lesions, etc.);lack of patient consent to participate in the study;absence of the patient′s neuroimaging archive in the follow-up.

### 2.3. Surgical Technique

All operations had the same surgical team. Patients were placed under intravenous anesthesia and artificial ventilation of the lungs. The patients were positioned prone with pads under their shoulders and iliac regions. Surgery was performed using a Pentero 900 operating microscope (Carl Zeiss, Berlin, Germany), Anspach Effort power equipment (Palm Beach Gardens, FL, USA), specialized instrumentation, and intraoperative fluoroscopy with Philips apparatus (Amsterdam, The Netherlands).

Patients were allocated into two groups. In the first group (*n* = 20), decompression/stabilization was performed using an Op-TLIF with a median approach and open transpedicular screw fixation (TSF) using the “Conmet” system (Moscow, Russia) (Figure 2); in the second group (*n* = 60), an MI-TLIF technique was used with an intermuscular paramedian approach and various minimally invasive dorsal rigid stabilization techniques. Patients of both groups underwent unilateral facetectomy, discectomy and foraminotomy for the spinal root, transforaminal installation of a cage made of PEEK material (“Pezo-T” (Ulrich Medical GmbH, Ulm, Germany), “T-pal” (Synthes, Solothurn, Switzerland), “Capstone” (Medtronic, Minneapolis, MN, USA)). Depending on the type of MI stabilization, three subgroups of patients were allocated: IIa (*n* = 20)—a rigid interspinous stabilizer “Coflex-F” (Paradigm Spine GmbH, Ulm, Germany) was used (Figure 3); IIb (*n* = 20)—the system of unilateral TSF “Viper II” (Synthes, Solothurn, Switzerland) and titanium facet cage “Facet Wedge” (Synthes, Solothurn, Switzerland) were implanted from the contralateral side (Figure 4); IIc (*n* = 20)—bilateral percutaneous TSF "U-centum" (Ulrich Medical GmbH, Ulm, Germany) was performed (Figure 5).

### 2.4. MR Imaging and Analysis

Evaluation of patient MRIs before and after surgery was conducted on sagittal and axial T2 weighted (T2w) images from a Siemens Magnetom Essenza MRI (1.5 T, Erlangen, Germany). Images were acquired during a single excitation with an echo time (TE) of 89 ms, a repetition time (TR) of 2500 ms, a matrix size of 320 × 256 with 4 mm slice thickness, and a field of view of 300 mm for sagittal T2w images. Images were acquired during a single excitation with a TE of 88 ms, a TR of 5784.8 ms, a matrix size of 320 × 240 with 4 mm slice thickness, and a field of view of 230 mm for axial T2w images. Analysis of de-identified JPEG images was performed independently by two specialists (one neurosurgeon and one radiologist) according to the Jarvik J. 2000 classification [12] at the operative level and adjacent segment IVDs: grade 1—marked blurring without definable margins; grade 2—blurring but definable margins; grade 3—minimal blurring; grade 4—sharp definition. The analysis included evaluating the central canal, dural sac, interbody space at the operative level, and left and right foramen. As specified by the Jarvik J. 2000 [12] classification method, three axial MRI slices per disc space were evaluated at the operative level and at two adjacent levels. Radiological grading [13] (grade 1—no significant artifact, grade 2—mild artifact measuring 1 mm or less surrounding the implant; grade 3—moderate artifact measuring greater than 1 mm but less than 3 mm; grade 4—severe artifact measuring greater than 3 mm surrounding the implant) and orthopedic grading [13] (grade 1—no reduction in diagnostic quality because of artifact, grade 2—some artifact with a reduction in diagnostic quality; grade 3—severe artifact with loss of diagnostic ability) scales were also used to assess the severity of MRI artifacts for the operative and adjacent segments [13]. As specified by the radiological and orthopedic grading, three axial MRI slices per disc space were evaluated at the operative level and at two adjacent ones. Kappa statistics (GraphPad Software, Inc., San Diego, CA, USA) were used to evaluate the interobserver agreement. The multifidus muscle area was calculated from anonymized axial T2w MRI images using the MultiVox DICOM Viewer software (Gamma Multivox, Moscow, Russia). Multifidus muscles were segmented from preoperative and postoperative MRIs using anatomical landmarks, and the total area for the right and left sides of each level were recorded (Figure 6a–c). The average muscle area across all three levels was calculated. The postoperative muscle areas were subtracted from the preoperative area for each patient, then divided by the preoperative area in order to assess a percent area change for each individual. Statistical analysis was performed on the percent area change metric as one indication of muscle atrophy. In this study, multifidus fat content was not assessed.

### 2.5. Statistical Analysis

Statistics were performed using Statistica software version 6.0 (StatSoft Inc., Tulsa, OK, USA). Descriptive statistics are presented as M ± SD, where M is the mean value and SD is the standard deviation. Comparison of continuous variables in the groups of respondents was performed using one-way ANOVA as amended by Bonferroni. Comparison of categorical variables in all of the scoring systems was performed using a Kruskal–Wallis test. A *p* value of less than 0.05 was considered significant.

## 3. Results

Patient demographics are shown in Table 1. There were no differences in gender and age between the studied groups (*p* > 0.05). The average follow-up period was 79.5 months in group I and 78 months in group II.

The interobserver agreement was evaluated using Kappa statistics for all Jarvik J. 2000 scores [12] (Table 2). It was found that there was moderate–excellent agreement between observers for all scores.

When testing hypotheses about the equality of variances using the Levene test, the equality of the analyzed variances was established (*p* > 0.05). The parametric univariate analysis of variance revealed the presence of differences only in the mean values in the groups in the analysis of postoperative visualization quality according to the Jarvik J. 2000 [12] scores at the operated level (*p* < 0.05) (Table 3). Pairwise comparison of the quality of postoperative imaging using the Bonferroni correction in the subgroups of minimally invasive rigid stabilization (IIa and IIb subgroups) showed fewer MRI artifacts and a significantly greater deterioration in MR images after Op (group I) and MI bilateral TSF (group IIc): P_I_-P_IIa_ < 0.001, P_I_–P_IIb_ < 0.001, P_I_–P_IIc_ < 0.001, P_IIa_–P_IIb_ = 0.24, P_IIa_–P_IIc_ = 0.02, P_IIb_–P_IIc_ = 0.04. In the groups of patients operated on with transpedicular screw stabilization, a deterioration in the quality of MR images in the projection of the spinal root canals at the operative level was revealed.

The Kappa statistics showed moderate–excellent interobserver agreement for both the radiological and orthopedic scores for MRI artifact evaluation [13] (Table 4). When testing hypotheses about the equality of variances using the Levene test, the equality of the analyzed variances was established (*p* > 0.05). The parametric univariate analysis of variance revealed the presence of differences in the mean values in the groups in the analysis of the quality of MR images on the radiological and orthopedic scales [13] at the operated level (*p* < 0.05) (Table 5). Pairwise comparison of the quality of postoperative imaging using the Bonferroni correction in the subgroups of showed better image quality of the operated segment after minimally invasive rigid stabilization (IIa and IIb subgroups) and a significant deterioration in postoperative MR images after Op (group I) and MI bilateral TSF (group IIc): P_I_–P_IIa_ < 0.001, P_I_–P_IIb_ = 0.001, P_I_–P_IIc_ = 0.25, P_IIa_–P_IIb_ = 0.003, P_IIa_–P_IIc_ < 0.001, P_IIb_–P_IIc_ = 0.009—according to the radiological scale [13]; P_I_–P_IIa_ < 0.001, P_I_–P_IIb_ < 0.001, P_I_–P_IIc_ = 0.42, P_IIa_–P_IIb_ = 0.002, P_IIa_–P_IIc_ < 0.001, P_IIb_–P_IIc_ = 0.016—according to the orthopedic scale [13].

The Kappa statistics showed moderate–good interobserver agreement for average multifidus muscle area across all three levels: group I—0.81 ± 0.16 (0.66–0.96, 95% CI); group IIa—0.84 ± 0.12 (0.67–0.92, 95% CI); group IIb—0.80 ± 0.17 (0.71–0.98, 95% CI); group IIc—0.74 ± 0.13 (0.61–0.90, 95% CI).

A comparative assessment of the area of the multifidus muscle by MRI of the lumbar spine over time showed statistically significant changes after Op (group I) and MI bilateral TSF (group IIc).

When testing hypotheses about the equality of variances using the Levene test, the equality of the analyzed variances was established (*p* > 0.05). The parametric univariate analysis of variance revealed the presence of differences in the postoperative MRI of the multifidus area muscle (*p* < 0.05) (Table 6). Pairwise comparison of the quality of postoperative imaging using the Bonferroni correction in the subgroups of showed less muscle atrophy after minimally invasive rigid stabilization and a significant greater atrophy after open decompressive-stabilization interventions: P_I_–P_IIa_ < 0.001, P_I_–P_IIb_ = 0.001, P_I_–P_IIc_ = 0.012, P_IIa_–P_IIb_ = 0.031, P_IIa_–P_IIc_ = 0.012, P_IIb_–P_IIc_ = 0.025. Minimum muscle atrophy was recorded in subgroups IIa and IIb of MI rigid stabilization, while after the placement of a rigid interspinous implant, fewer changes were detected in the multifidus muscle compared with that in unilateral TSF and contralateral placement of a titanium facet cage.

## 4. Discussion

Rigid stabilization of the spine leads to significant biomechanical changes and overload of adjacent segments, accelerating their degeneration [1,5]. In addition, classical decompression and stabilization interventions are associated with substantial damage and atrophy of the multifidus muscle, which maintains the dynamic lumbar and spinal pelvic balance [14]. The use of MI technologies in spinal surgery reduces surgical trauma to the skin, paravertebral soft tissues, and bone structures [15]. Less-invasive surgeries have better long-term clinical results, primarily associated with preserving the multifidus muscle’s nerves and causing its minimal postoperative atrophy [16].

Analysis of MR images after rigid stabilization of the spine is necessary to monitor neural structures, paravertebral tissues, and adjacent segments [7]. Simultaneously, the type of implanted material significantly affects the postoperative imaging quality from minimal artifacts in titanium devices to much larger ones in stainless steel devices [17].

Dorsal decompression and stabilization interventions are performed on the anterior and posterior vertebral column using different fixing devices. For interbody implantation, cages are made primarily of polyether ether ketone (PEEK) or titanium [3], while stainless steel, titanium, cobalt–chromium–molybdenum alloy, or PEEK are used for dorsal stabilization [4]. The size of artifacts for these different materials can be estimated from their magnetic susceptibility coefficients (Table 7), where negative values indicate diamagnetic materials, near-zero values indicate paramagnetic materials, and large values indicate ferromagnetic materials [11,18].

Titanium interbody cages, having high strength and biocompatibility, are most often used to stabilize the anterior supporting complex, but they are paramagnetic and can distort the magnetic field and impede visualization of the anatomy of the operative area [19,20]. Polymer composites reinforced with carbon fiber or based on PEEK are common alternative nonmetallic biomaterials for interbody spacers [21,22]. In the first case, minimal artifacts are recorded on postoperative MRI [23]. The latter are diamagnetic and lack X-ray contrast, making it possible to both view them on an MRI and assess bone formation within the cage using X-ray [24]. In this study, we did not aim to analyze interbody implants’ effect on postoperative MRI quality. Therefore, only patients with PEEK transforaminal cages were included.

Op-TLIFs are often limited to the use of transpedicular screw fixation systems to stabilize the posterior support complex. In contrast, MI techniques can use a more comprehensive selection of implants, including interspinous spacers and transfacet stabilization [5]. Despite the different clinical and instrumental indications for using the implants mentioned above, the main goal of such techniques is complete decompression and effective stabilization, reducing the trauma of surgical intervention [2,3].

In the literature, few studies are devoted to analyzing MR image quality after dorsal decompression and stabilization interventions. Titanium screw and rods have lower magnetic susceptibility than stainless-steel and cobalt–chromium [25]. Thus, in the study of Ahmad et al. [6] a larger artifact size was observed on postoperative MRI in longitudinal rods made of cobalt–chromium material compared to titanium ones on T1w—11.8 ± 1.8 and 8.5 ± 1.2 mm, respectively, and on T2w—11.0 ± 2.3 and 8.3 ± 1.7 mm, respectively. In this case, the interpretation of the state of the operative area was not difficult. Cobalt–chromium implants are mechanically stronger than titanium ones [26], contributing to the widespread adoption of titanium–cobalt–chrome longitudinal rods and screw heads of smaller lengths and diameter for spinal surgery [27]. Trammell et al. [28] confirmed on cadaver material comparable neuroimaging data in the interpretation of the operated and adjacent segments when placing (1) 6.5 mm titanium screws with cobalt–chromium–molybdenum polyaxial heads and cobalt–chromium–molybdenum longitudinal rods with a diameter of 4.75 mm compared with (2) 6.5 mm titanium polyaxial screws and titanium longitudinal rods with a diameter of 5.5 mm.

There are several methods aimed at reducing the magnetic susceptibility artifacts of implants: the use of low-field [29] or high-field [30] MRI; control of MR scanning parameters—slice thickness, throughput, and echo time [31]; slice encoding for metal artifact correction (SEMAC) [32]; technologies of iterative decomposition of water and fat with echo asymmetry and least-squares estimation, IDEAL [33]. The disadvantages of these techniques are the deterioration of the quality of postoperative MR imaging of soft tissues, prolongation of the scan time, the need for additional expensive modules in the MRI apparatus, and radiologists’ training to work with these protocols [29,30,31,32,33].

This study has shown that using various types of implants to stabilize the posterior support complex during single-level MI-TLIF does not impair the quality of postoperative visualization of the adjacent segments. It was found that MI and Op bilateral TSF have more significant artifacts on MRI, making it difficult to interpret the spinal root canals’ state at the operative level and objectively assess the multifidus muscle area. It was also found that implants used in the MI-TLIF group have lower magnetic susceptibility and atrophic changes in the multifidus muscle compared to open placement of transpedicular screw systems. In our opinion, the increase in postoperative MRI quality and a decrease in atrophic changes in the multifidus muscle can be attributed to both decreased surgical trauma and less hardware stabilizing the posterior support complex of implants. Thus, the search for device materials and ways to improve postoperative imaging to diagnose structural malfunctions, diseases of adjacent segments, and atrophic changes in the multifidus muscle are currently ongoing.

Analysis of indications for a particular type of surgical intervention, its technical component, and neurological outcomes are beyond the scope of this study, which aimed exclusively to compare MR characteristics of various devices for MI single-segment dorsal rigid stabilization and Op rigid fusion. However, the study of clinical data after low-traumatic placement of these structures in previously published papers did not reveal significant intergroup differences. Additionally, the best long-term outcomes were recorded after MI-TLIF in terms of pain level assessed using a visual analog scale, functional status according to ODI and SF-36 quality of life scale compared with Op-TLIF.

In our opinion, the results obtained by us on which consumable will cause a greater artifact in postoperative MRI should not solely influence the choice for a particular consumable. Other important factors for spinal surgeons in considering different implants are the possibility of MI placement, wear resistance, speed of integration and formation of the bone block, as well as the ease of implantation and revision. Nevertheless, we do not exclude the possibility of choosing, under all equal conditions, the intervention with implants that reduce the number of artifacts in postoperative MRIs and, thereby, make the visualized postoperative outcome more obvious.

### 4.1. Limitations

The limitations of the study, potentially having the ability to influence its results, should include (1) low number of patients in each group; (2) retrospective study design; (3) nonrandomized sample of examined patients without using an electronic patient data management system; (4) the use of multicomponent structurally different implants; (5) the use of 1.5 T MRI and T2w imaging exclusively without additional software modules; (6) lack of analysis of the influence of other factors that can cause the presence of MR artifacts; (7) the use of percent change of muscle area as the sole factor in assessing muscle atrophy.

### 4.2. Strengths of the Study

This study has a number of strengths: (1) the constructs used have not been previously studied for individual and comparative analysis of the quality of postoperative images; (2) limiting the level of operation of L4–L5 segments ensured homogeneity of study groups in the interpretation of anatomical changes in the operated area; (3) in each of the three groups, different parameters of MR images were blindly evaluated by two experts in all patients (*n* = 60); (4) the use of a complex of MRI characteristics (classification by Jarvik J., [12] radiological and orthopedic scales for assessing artifacts [13], calculation of the area of a multifidus muscle) after minimally invasive single-level dorsal rigid stabilization has not previously been used.

## 5. Conclusions

Postoperative MRI for assessing neural structures, paravertebral tissues, and segments adjacent to the operation depend on both the surgical intervention and type of structure used to stabilize the posterior support complex. The techniques of Op and MI bilateral TSF have a comparable deterioration in visualization of the operated segment with significantly greater atrophy of the multifidus muscle after performing the Op-TLIF technique. Op and MI single-segment dorsal rigid stabilization does not reduce the possibility of adequate postoperative MRI assessment of levels adjacent to the operation. MI-TLIF has lower magnetic susceptibility artifacts and change in multifidus muscle area, which gives the ability to study the postoperative volume of decompression and atrophy of the paravertebral muscles in the follow-up period. Additionally, a rigid interspinous implant has minimal postoperative artifacts compared to unilateral TSF with a contralateral placement of a titanium facet cage and bilateral TSF.

## Figures and Tables

**Figure 1 diagnostics-11-00906-f001:**
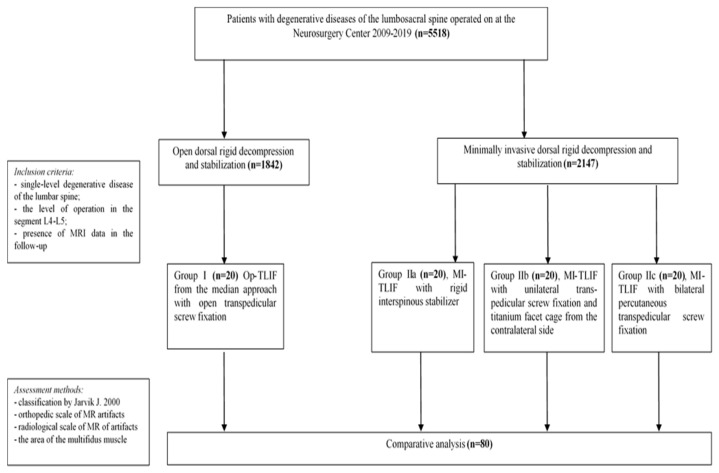
Flow chart characterizing the study design.

**Figure 2 diagnostics-11-00906-f002:**
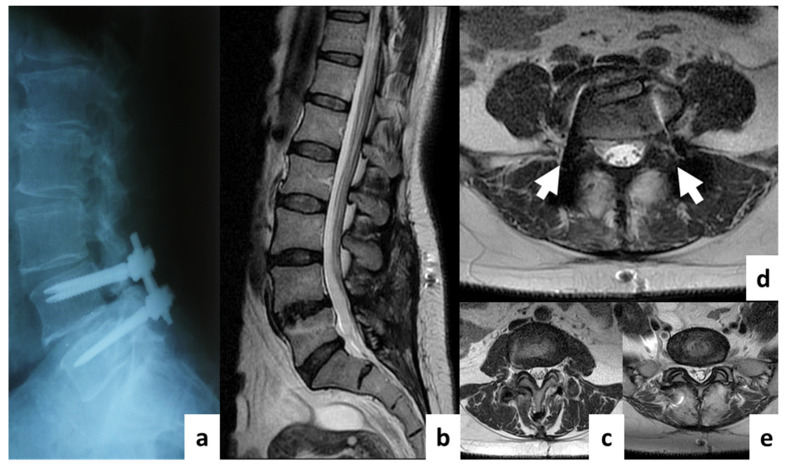
Postoperative T2w MRI of a 50-year-old female from group I with degenerative disc disease at L4–L5. (**a**)—sagittal X-ray; (**b**)—sagittal MRI; (**c**)—axial MRI at L3–L4 (overlying segment); (**d**)—axial MRI at L4–L5 (operation level); (**e**)—axial MRI at L5–S1 (underlying segment). The white arrows indicate regions with visible artifacts.

**Figure 3 diagnostics-11-00906-f003:**
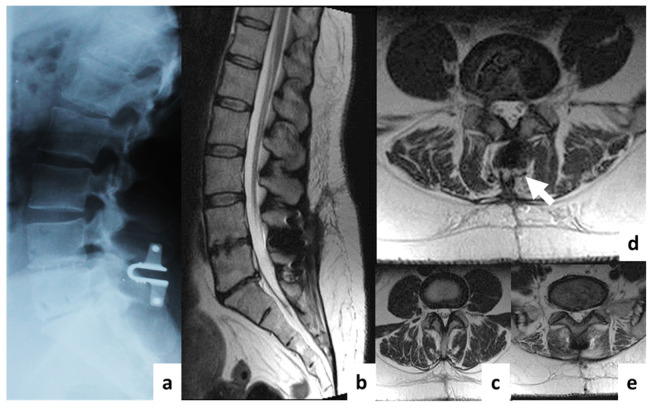
Postoperative T2w MRI of a 44-year-old male from group IIa with degenerative disc disease at L4–L5. (**a**)—sagittal X-ray; (**b**)—sagittal MRI; (**c**)—axial MRI at L3–L4 (overlying segment); (**d**)—axial MRI at L4–L5 (operation level); (**e**)—axial MRI at L5–S1 (underlying segment). The white arrows indicate regions with visible artifacts.

**Figure 4 diagnostics-11-00906-f004:**
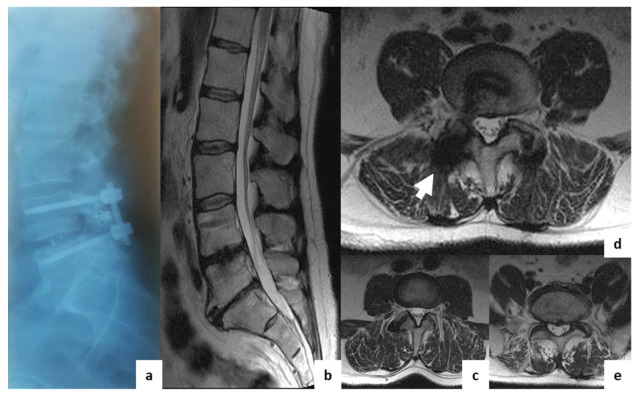
Postoperative T2w MRI of a 46-year-old female from group IIb with degenerative disc disease at L4–L5. (**a**)—sagittal X-ray; (**b**)—sagittal MRI; (**c**)—axial MRI at L3–L4 (overlying segment); (**d**)—axial MRI at L4–L5 (operation level); (**e**)—axial MRI at L5–S1 (underlying segment). The white arrows indicate regions with visible artifacts.

**Figure 5 diagnostics-11-00906-f005:**
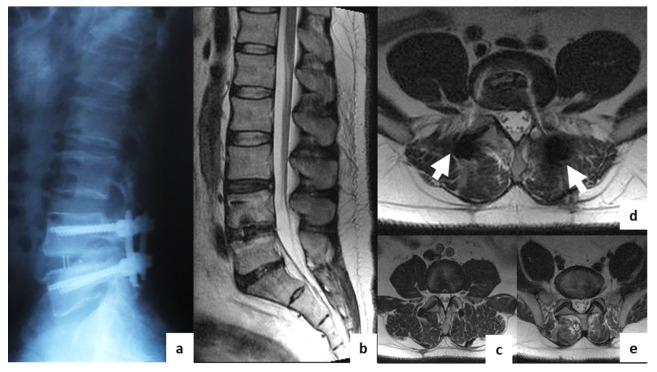
Postoperative T2w MRI of a 43-year-old male from group IIc with degenerative disc disease at L4–L5. (**a**)—sagittal X-ray; (**b**)—sagittal MRI; (**c**)—axial MRI at L3–L4 (overlying segment); (**d**)—axial MRI at L4–L5 (operation level); (**e**)—axial MRI at L5–S1 (underlying segment). The white arrows indicate regions with visible artifacts.

**Figure 6 diagnostics-11-00906-f006:**
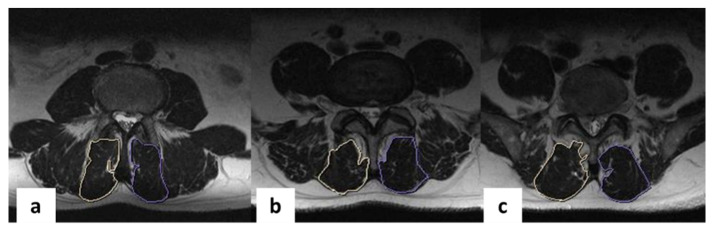
Axial MRI scans of the lumbar spine, the right multifidus muscle is outlined in white, the left multifidus muscle is outlined in purple: (**a**)—L3–L4; (**b**)—L4–L5; (**c**)—L5–S1.

**Table 1 diagnostics-11-00906-t001:** Patient demographic information.

Criteria	Group I (*n* = 20)	Group II (*n* = 60)			F	*p*
		Subgroup IIa (*n* = 20)	Subgroup IIb (*n* = 20)	Subgroup IIc (*n* = 20)		
Age, years	46.4 ± 5.7	44.5 ± 3.9	43.4 ± 6.2	47.4 ± 7.3	0.31	0.63
Male/female ratio, *n* (%)	11 (55)/9 (45)	12 (60)/8 (40)	13 (65)/7 (35)	10 (50)/10 (50)	0.25	0.85
Observation period, months	79.5 ± 2.54	78.1 ± 1.85	77.0 ± 0.81	78.3 ± 3.23	0.88	0.45

**Table 2 diagnostics-11-00906-t002:** Interobserver agreement of Jarvik 2000 [12] scores.

Criteria	Group I (*n* = 20)		Group II (*n* = 60)					
			Subgroup IIa, *n* = 20		Subgroup IIb, *n* = 20		Subgroup IIc, *n* = 20	
	Kappa ± SE	95% Confidence Interval	Kappa ± SE	95% Confidence Interval	Kappa ± SE	95% Confidence Interval	Kappa ± SE	95% Confidence Interval
Dural sac at operation level	0.86 ± 0.15	0.74–0.93	0.95 ± 0.04	0.85–1.00	0.75 ± 0.13	0.69–0.90	0.85 ± 0.14	0.76–0.94
Interbody space at operation level	0.79 ± 0.12	0.66–0.91	0.81 ± 0.11	0.68–0.92	0.80 ± 0.11	0.77–0.92	0.95 ± 0.14	0.87–1.00
Central canal at operation level	0.80 ± 0.12	0.77–0.93	0.85 ± 0.10	0.64–0.95	0.74 ± 0.14	0.66–0.84	0.95 ± 0.04	0.85–1.00
Right foramen	0.95 ± 0.04	0.85–1.00	0.80 ± 0.09	0.63–0.92	0.85 ± 0.14	0.76–0.94	0.75 ± 0.10	0.64–0.90
Left foramen	0.85 ± 0.10	0.65–0.92	0.80 ± 0.10	0.66–0,92	0.77 ± 0.15	0.65–0.85	0.80 ± 0.11	0.77–0.92
Upper adjacent level	0.72 ± 0.12	0.61–0.85	0.75 ± 0.10	0.64–0.90	0.85 ± 0.13	0.79–0.90	0.81 ± 0.11	0.68–0.92
Lower adjacent level	0.85 ± 0.14	0.76–0.94	0.85 ± 0.10	0.64–0.95	0.95 ± 0.14	0.87–.00	0.72 ± 0.12	0.61–0.85

**Table 3 diagnostics-11-00906-t003:** Comparison of the image quality according to the Jarvik 2000 [12] scores.

Criteria	Preoperative						Postoperative					
	Group I (*n* = 20)	Group IIa (*n* = 20)	Group IIb (*n* = 20)	Group IIc (*n* = 20)	F	*p*	Group I (*n* = 20)	Group IIa (*n* = 20)	Group IIb (*n* = 20)	Group IIc (*n* = 20)	F	*p*
Operation level	3.96 ± 0.12	3.96 ± 0.17	3.97 ± 0.17	3.98 ± 0.17	0.20	0.88	2.64 ± 0.29	3.94 ± 0.17	3.11 ± 0.12	2.96 ± 0.26	*41.63*	<*0.001*
Adjacent level	3.98 ± 0.10	3.98 ± 0.15	3.98 ± 0.16	3.97 ± 0.16	0.53	0.66	3.62 ± 0.23	3.89 ± 0.12	3.87 ± 0.27	3.65 ± 0.46	0.34	0.61
Overall score	3.94 ± 0.36	3.97 ± 0.14	3.97 ± 0.14	3.97 ± 0.15	0.18	0.91	3.17 ± 0.26	3.91 ± 0.12	3.59 ± 0.16	3.34 ± 0.27	0.42	0.74

Note: *p*-values were calculated using one-way ANOVAs and indicate significance between the four groups at their respective time points; bold for emphasis—the presence of statistically significant intergroup differences.

**Table 4 diagnostics-11-00906-t004:** Interobserver agreement by radiological and orthopedic scales for MRI artifact evaluation [13].

Criteria	Group I (*n* = 20)		Group II (*n* = 60)					
			Subgroup IIa, *n* = 20		Subgroup IIb, *n* = 20		Subgroup IIc, *n* = 20	
	Kappa ± SE	95% Confidence Interval	Kappa ± SE	95% Confidence Interval	Kappa ± SE	95% Confidence Interval	Kappa ± SE	95% Confidence Interval
Radiology scale: operation level	0.86 ± 0.07	0.67–1.00	0.95 ± 0.10	0.78–1.00	0.85 ± 0.14	0.81–1.00	0.91 ± 0.12	0.87–1.00
Radiology scale: upper adjacent level	0.90 ± 0.16	0.82–1.00	0.85 ± 0.08	0.68–1.00	0.76 ± 0.10	0.68–0.95	0.84 ± 0.11	0.67–0.93
Radiology scale: lower adjacent level	0.78 ± 0.17	0.70–0.91	0.82 ± 0.10	0.66–0.95	0.73 ± 0.14	0.67–0.82	0.78 ± 0.15	0.69–0.83
Orthopedic scale: operation level	0.82 ± 0.10	0.66–0.92	0.80 ± 0.09	0.71–0.98	0.92 ± 0.15	0.85–1.00	0.90 ± 0.18	0.80–0.99
Orthopedic scale: upper adjacent level	0.81 ± 0.10	0.72–0.94	0.85 ± 0.08	0.68–1.00	0.75 ± 0.10	0.64–0.95	0.72 ± 0.16	0.61–0.92
Orthopedic scale: lower adjacent level	0.75 ± 0.20	0.67–0.92	0.75 ± 0.10	0.64–0.92	0.80 ± 0.14	0.71–0.91	0.79 ± 0.19	0.70–0.90

**Table 5 diagnostics-11-00906-t005:** Comparative assessment of the quality of postoperative MRI [13].

Criteria	Group I (*n* = 20)	Group II (*n* = 60)			F	*p*
		Subgroup IIa (*n* = 20)	Subgroup IIb (*n* = 20)	Subgroup IIc (*n* = 20)		
Radiology scale: operation level	2.83 ± 0.37	1.11 ± 0.30	1.99 ± 0.40	2.95 ± 0.48	*37.17*	<*0.001*
Radiology scale: upper adjacent level	1.21 ± 0.31	1.04 ± 0.22	1.08 ± 0.24	1.15 ± 0.36	0.76	0.23
Radiology scale: lower adjacent level	1.13 ± 0.34	1.15 ± 0.33	1.16 ± 0.43	1.17 ± 0.47	0.88	0.14
Orthopedic scale: operation level	2.39 ± 0.52	1.13 ± 0.30	2.04 ± 0.30	2.50 ± 0.50	*14.49*	<*0.001*
Orthopedic scale: upper adjacent level	1.25 ± 0.36	1.05 ± 0.22	1.02 ± 0.24	1.23 ± 0.47	0.63	0.34
Orthopedic scale: lower adjacent level	1.21 ± 0.72	1.03 ± 0.33	1.05 ± 0.41	1.10 ± 0.77	0.51	0.63

Note: *p*-values were calculated using one-way ANOVAs and indicate significance between the four groups within their respective scoring system; bold for emphasis—the presence of statistically significant intergroup differences.

**Table 6 diagnostics-11-00906-t006:** Changes of the multifidus muscle of the studied group of patients.

Criteria	Group I (*n* = 20)			Group II (*n* = 60)									F	*p*	F	*p*
				Subgroup IIa, *n* = 20			Subgroup IIb, *n* = 20			Subgroup IIc, *n* = 20						
	Preoperative	Postoperative	Mean Changes, %	Preoperative	Postoperative	Mean Changes, %	Preoperative	Postoperative	Mean Changes, %	Preoperative	Postoperative	Mean Changes, %	Preoperative		Postoperative	
Average multifidus muscle area, mm^2^	6.6 ± 1.2	3.0 ± 1.5 ***	54.5	6.3 ± 0.7	6.1 ± 1.6	3.2	6.4 ± 1.2	5.8 ± 1.1	9.4	6.4 ± 1.9	5.0 ± 1.4 ***	21.9	1.74	0.17	*23.82*	<*0.001*

Note: *—shows values that have statistically significant differences, bold for emphasis—the presence of statistically significant intergroup differences. *p*-values were calculated using one-way ANOVAs and indicate significance between the four groups on the percent changes at their respective time points.

**Table 7 diagnostics-11-00906-t007:** Comparison of the magnetic susceptibility coefficients of common medical materials.

Material	Magnetic Susceptibility (10^−6^ cm^3^ g^−1^)
18Cr–14Ni–2.5Mo	36.1 [11]
Co–18Cr–6Mo	8.37 [18]
Ti–6Al–4V	4.52 [18]
Ti–Gr2	2.43 [11]
Zr–1Mo	1.05 [11]

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
