# Peer review of "Comparison of MRI Visualization Following Minimally Invasive and Open TLIF: A Retrospective Single-Center Study"

_diagnostics, 2021, doi:10.3390/diagnostics11050906_

Round 1
Reviewer 1 Report
The authors analyse the quality of MRI examinations after transforaminal interbody fusion and stabilization. 80 patients receiving four different types of implants were evaluated before and after surgery. 20 patients, respectively, received either TLIF and open dorsal transpedicular srew fixation (TSF), TLIF and an interspinous spacer, TLIF and unilateral TSF and a facet cage, or TLIF and bilateral percutaneous TSF, respectively. MRIs before and after surgery were analysed for artefacts by two persons and image quality graded according to Jarvik, a radiological grading and an orthopedic grading. In addition, the area of the multifidus muscle was quantified. The authors conclude that TLIF and open TSD and TLIF and blilateral percutaneous TSF have the same reduced quality of postoperative MRI which is inferior to the other two groups. The adjacent segments were not affected.
Comments:
How were the patients selected? Was the patient identification process based on an automatized search using an electronic patient data management system? Who picked the patients? When patients were included that had post-surgical imaging within 5 and 8 years after surgery than surgery was performed not later than 2014 and not 2019 as indicated.
How many MRI-slices were evaluated per segment (one over several images). Were sagittal and axial images analysed or only axial images.
Figures 2 to 5: Images are too small. Nothing can be seen from the axial images.
Statistical analysis: The authors use a grading system. The calculation of means in a grading system makes no sense (or is grade four twice as worse or twice as better than grade 2?).
What kind of specialists did evaluate the images? Neurosurgeons, orthopedics, radiologists?
The evaluation cannot be blinded, as the type of implant can be seen on the images.
How did the authors define the muscle atrophy? This subject is more complex and many other co-factors should be recognized for a sufficient evaluation of muscle atrophy, such like physical fitness of patients, BMI etc..
Clinical data are not within the scope of this study. Nevertheless, the mean patients’ age in the cohort of the study is relatively low (48 years) for patients with degenerative lumbar diseases who need TLIF surgery. How can the authors explain this? Which criteria indicate a TLIF approach in their local work-up algorithm?
Table 3, table 5, table 6: To what does the p-level refer to?
CT scans for the illustrative cases might be also helpful to demonstrate the location of the implants.
Further limitations of the study are the low number of patients in each gropu, the way to select the patients (not really explained) from the database, the retrospective design, the evaluation that cannot be blinded as the type of implant can be seen on the image (e.g. interspinous device, contralateral facet cage, bilateral TSF).
Reviewer 2 Report
The authors conducted a retrospective single centre study in order to compare MRI visualization following Minimal Invasive and Open-transforaminal interbody fusion.
The originality of the study seems sufficient, as comparative analysis of MRI artifacts after Op and MI one-segment decompression and stabilization interventions with implantation of structurally different systems has not been performed before.
The Introduction section might be shortened, as the 2nd para of the Introduction is not related to the topic being discussed (lines 51-65).
Materials and Methods are sufficiently descriptive, including the Flow chart characterizing the study design. Patient inclusion and exclusion criteria are presented. Surgical technique is discussed.
Only MRI techniques is described in the section 2.4 Study outcomes. They should be included in the Methods section.
The Results section is wide enough, including comparative tables.
Round 2
Reviewer 1 Report
The questions and comments were adequately considered and the manuscript was changed accordingly. There are no further comments.